Application of the urban exposome framework using drinking water and quality of life indicators: a proof-of-concept study in Limassol, Cyprus

http://orcid.org/0000-0002-2906-5743 Andrianou Xanthi D. 1
van der Lek Chava 1 2
http://orcid.org/0000-0001-7260-192X Charisiadis Pantelis 1
Ioannou Solomon 1
Fotopoulou Kalliopi N. 1
Papapanagiotou Zoe 3
Botsaris George 3
Beumer Carijn 2
Makris Konstantinos C. 1 konstantinos.makris@cut.ac.cy
1 Cyprus International Institute for Environmental and Public Health, Cyprus University of Technology , Limassol , Cyprus
2 Department of Health, Ethics and Society, Faculty of Health, Maastricht University , Maastricht , Netherlands
3 Department of Agricultural Sciences, Biotechnology and Food Sciences, Cyprus University of Technology , Limassol , Cyprus
Nock Nora
Electronic publication date: 2019 May 24
Publication date: 2019
Volume: 7
Electronic Location ID: e6851
Received 2018 Dec 10; Accepted 2019 Mar 26
Copyright: © 2019 Andrianou et al.
Copyright year: 2019
Copyright holder: Andrianou et al.
License: This is an open access article distributed under the terms of the Creative Commons Attribution License, which permits unrestricted use, distribution, reproduction and adaptation in any medium and for any purpose provided that it is properly attributed. For attribution, the original author(s), title, publication source (PeerJ) and either DOI or URL of the article must be cited.
License URL: https://creativecommons.org/licenses/by/4.0/

Keywords: Exposome, Urban health, Small area, Epidemiology

Funding: Internal funding provided by Konstantinos C. Makris The study was conducted with internal funding provided by Konstantinos C. Makris. The funders had no role in study design, data collection and analysis, decision to publish, or preparation of the manuscript.

==============================
Background

Cities face rapid changes leading to increasing inequalities and emerging public health issues that require cost-effective interventions. The urban exposome concept refers to the continuous monitoring of urban environmental and health indicators using the city and smaller intra-city areas as measurement units in an interdisciplinary approach that combines qualitative and quantitative methods from social sciences, to epidemiology and exposure assessment.

Methods

In this proof of concept study, drinking water and quality of life indicators were described as part of the development of the urban exposome of Limassol (Cyprus) and were combined with agnostic environment-wide association analysis. This study was conducted as a two-part project with a qualitative part assessing the perceptions of city stakeholders, and quantitative part using a cross-sectional study design (an urban population study). We mapped the water quality parameters and participants’ opinions on city life (i.e., neighborhood life, health care, and green space access) using quarters (small administrative areas) as the reference unit of the city. In an exploratory, agnostic, environment-wide association study analysis, we used all variables (questionnaire responses and water quality metrics) to describe correlations between them.

Results

Overall, urban drinking-water quality using conventional indicators of chemical (disinfection byproducts-trihalomethanes (THM)) and microbial (coliforms, E. coli, and Enterococci) quality did not raise particular concerns. The general health and chronic health status of the urban participants were significantly (false discovery rate corrected p-value < 0.1) associated with different health conditions such as hypertension and asthma, as well as having financial issues in access to dental care. Additionally, correlations between THM exposures and participant behavioral characteristics (e.g., household cleaning, drinking water habits) were documented.

Conclusion

This proof-of-concept study showed the potential of using integrative approaches to develop urban exposomic profiles and identifying within-city differences in environmental and health indicators. The characterization of the urban exposome of Limassol will be expanded via the inclusion of biomonitoring tools and untargeted metabolomics.

Introduction

The definition of the exposome in 2005 by Dr. Wild introduced a paradigm in environmental and population health research, which promotes studies that either encompass simultaneous assessment of multiple exposures of the general population or focus on specific time windows of susceptibility (e.g., pregnancy), to capture the totality of environmental/lifestyle/behavioral exposures (Wild, 2005; Robinson et al., 2015; Andra et al., 2015; Cui et al., 2016). The definition of the exposome along with the advances in methodologies for high throughput analysis in shorter time has also redefined the study paradigm in environmental health (Buck Louis, Smarr & Patel, 2017). Thus, decoding the exposome will not benefit only environmental health, but it will lead to better understanding of disease development and progression and it will allow for comprehensive monitoring of environment-disease associations.

The exposome as a paradigm has fostered innovation in exposure assessment. It allows for intra- and inter-disciplinary approaches in public health to become more widespread than they are now, as it is indicated by the number of “exposomes” that have been defined to address different totalities and with different units of reference (National Research Council, 2012; Escher et al., 2016; Dai et al., 2017).

Cities are dynamic and complex systems that become the future focus of public health systems, because they currently host more than half of the global population and generate >80% of the global GDP (Urban Development Overview, 2018; Zhang, 2011). Using terminology similar to the one used for the human exposome, the urban exposome extends the utility of the human exposome, and it is defined as the totality of indicators (quantitative or qualitative) that shape the quality of life and health of urban populations (Andrianou & Makris, 2018). Monitoring of these indicators that can be either external or internal city parameters, is not merely the sum of individual human exposures, but places cities and their smaller areas in the center of an urban-oriented study framework, previously defined as the urban exposome framework (Andrianou & Makris, 2018). Following the urban exposome approach, the quality of life in urban centers is concurrently assessed together with other indicators, such as water quality, or prevalence/incidence of communicable, and non-communicable diseases using interdisciplinary methodology. This framework expands previous study approaches where the urban exposome is defined as the sum of exposures in cities (Probst-Hensch, 2017; Robinson et al., 2018), and proposes a more interdisciplinary and holistic approach to assess health determinants of the general population.

To present the first application of the urban exposome framework, we used drinking-water quality metrics and quality of life indicators. As more people nowadays live in cities, providing easy access to safe and affordable water, as well as eliminating possible within-city health and societal disparities becomes more challenging. Besides the technical provisions to maintain water of good quality, the uninterrupted availability of water becomes an issue due to climate change manifestations, especially in areas that are already or expected to be hit harder by extended droughts and other related meteorological phenomena. Europe, overall, is expected to face increases both in the extent of geographical areas affected by droughts and in the duration of such climatic events (Samaniego et al., 2018). In particular, cities located in Southeast Europe and the Mediterranean region are predicted to face challenges in maintaining water availability and adequate water quality in the future (Samaniego et al., 2018). Therefore, cities and their smaller areas (i.e., neighborhoods, or other small administrative areas) are warranted to address water-related issues, such as water demand, safety, security, and quality issues, while tackling societal inequalities and health disparities. In this context, the concept and the study framework of the urban exposome can be introduced to help scientists and policy makers in the systematic spatio-temporal surveillance and monitoring of a city’s heterogeneous health profile.

The urban exposome study framework has a hypothesis-generating scope and goes beyond the classical one exposure-one outcome approach urban studies. Therefore, our aim was to present a proof-of-concept study using the urban exposome framework to monitor urban health dynamics. The study setting was Limassol, Cyprus. The urban center of Limassol (“city of Limassol”) is defined as a medium-sized city (∼200,000 inhabitants according to the 2011 Population Census of Cyprus) (Ministry of Finance of the Republic of Cyprus, 2014; Eurostat, 2017) and it consists of the municipality of Limassol and neighboring municipalities. The city of Limassol currently faces rapid economic development with half of the urban population of the city residing within the municipality of Limassol (∼110,000 inhabitants).

The specific objective of this analysis was to describe chemical and microbiological parameters of drinking water quality, coupled with quality of life indicators as measured in the municipality of Limassol, in summer 2017. In this approach we used quarters which are the smallest within-municipality administrative areas, as the unit of reference. To complete the urban exposome of Limassol to the extent possible, additional analysis of routine surveillance data is necessary but it goes beyond the scope of the present analysis.

Materials and Methods

Application of the urban exposome in the municipality of Limassol, Cyprus

We have previously described the urban exposome as all indicators that need to be continuously monitored for the assessment of city health (Andrianou & Makris, 2018). Within the framework of the urban exposome, external to the city parameters that cannot be influenced by the city itself can be either general (e.g., global trends and policy decisions) or more specific (e.g., climate change impacts, demographic changes, culture). It follows that internal parameters are those that are integral to the city, such as infrastructure, built/neighborhood environment and determinants of population health (e.g., socioeconomic factors). Either external or internal to the city, the parameters described before can influence and be influenced by each other both horizontally (within each domain) and/or vertically (between domains). A study was conducted in the summer of 2017 to describe the water and quality of life aspects of the urban exposome of Limassol, following an integrative and interdisciplinary approach (Fig. 1). This approach included the following parts:A perceptions survey with a mixed-methods approach to evaluate the perceptions of stakeholders (i.e., citizens and municipality officials) about the quality of life and certain environmental risks (e.g., chemical and microbial risks in drinking-water) in the city.

A cross-sectional urban population study with a short questionnaire and collection of tap water samples from households distributed in the quarters of the municipality of Limassol to evaluate water quality indicators and citizens’ attitudes about the environment, quality of life in the city and self-reported health status.

Figure 1 Urban exposome—human exposome continuum, and the practical application of the urban exposome framework in the urban setting of Limassol city.

The parts of the urban exposome specifically discussed in the current analysis include a perceptions study and an urban population study, which includes parameters measured in drinking-water to assess the water quality coupled with questionnaire responses about individual lifestyle, behavioral, and personal health indicators.

In this analysis, we focus on the general external urban exposome through the perceptions study and the internal exposome domain through the inclusion of drinking water chemical/microbiological indicators, responses on neighborhood’s quality of life and participant characteristics (Fig. 1). Using the human exposome framework as a reference, this study also explored the human exposome domains, that is, the general external domain (perceptions study), the specific external domain (drinking water and quality of life indicators) and the internal domain (participant characteristics) (Fig. 1).

Perceptions study

For the perceptions study, the urban community-based participatory research methodology was applied to actively engage the municipality of Limassol and citizens (identified as community stakeholders) about urban health issues (Rojas & Neutra, 2008). The stakeholders’ perceptions were assessed using a mixed methods approach. Face-to-face interviews were conducted, and short questionnaires were administered to the municipality officials (i.e., technical officers from the municipality of Limassol and neighboring municipalities). The questions asked during the face-to-face interviews and the questionnaire for the municipality officers focused on the identification of trends shaping the city of Limassol, assessment of the climate change manifestations and their impact, scoring of environmental health concerns, an assessment of what was believed to be the citizens’ major health threats with regards to urban life, and perspectives about future opportunities to improve health in the city. For the citizens’ perceptions, an online anonymous questionnaire initially distributed via email among staff at the Cyprus University of Technology campus that is in the municipality of Limassol, and then to the general public via mailing lists maintained at the Cyprus International Institute for Environmental and Public Health for the dissemination of newsletters, and social media (Facebook). This online questionnaire included various questions on climate change perceptions, environmental concerns in general, health perceptions, and perceptions about drinking-water quality.

Urban population study

A cross-sectional population study was set up in the municipality of Limassol, Cyprus. Participants (n = 132) were recruited after being informed about the study through phone calls made in collaboration with the municipality of Limassol, in the summer of 2017 from all quarters of the municipality, following the 2011 Census population distribution (Table S1). For this study, we also collected urine samples which will be used in the biomonitoring part (measurement of chemical exposures in urine) of the assessment of the urban exposome (not included in this analysis). Thus, sample size estimations were based on the assumption that a total of 120 participants randomly selected from the whole municipality would allow us to evaluate the baseline levels of environmental exposures, as according to previous biomonitoring studies a sample of at least 120 randomly selected participants is adequate to capture the 95th percentile of the population levels (Becker et al., 2013). Quarters that are small in area and population, with one participant, were merged with neighboring ones and three quarters located along the beachfront, each having one participant were also merged together (Fig. S1). To ensure high spatial coverage, only one participant was recruited per street.

Tap water samples were collected from all participating households and in situ measurements of free chlorine were taken during house visits. The drinking water indicators that were assessed were trihalomethanes (THM) along with the free chlorine, from the category of the chemical parameters, and total coliforms, E. coli, Enterococcus spp., Pseudomonas aeruginosa, and total viable counts (TVC) at 22 and 37 °C, from the category of the microbial parameters. These indicators (besides free chlorine) are also routinely monitored in the European Union (EU Council, 1998). All participants were asked to complete a questionnaire that included, among others, questions about life in their neighborhoods, self-reported health status, and drinking water habits. The questionnaire was adapted by the validated urban health and the European Health Survey questionnaires (Republic of Cyprus, Ministry of Finance, 2014; European Urban Health Indications System Part 2, 2012; Pope et al., 2016). From this questionnaire the quality of life indicators (e.g., access to health care services, life in the neighborhood and use of green spaces) were assessed.

The study was approved by the National Bioethics Committee of Cyprus (decision number: 2017/23). All participants read and signed the informed consent documents before data collection. The STROBE statement is available in the Supplementary Material (von Elm et al., 2007).

Water sampling and analysis

The main faucet used for satisfying the water needs of the participating household was selected for drinking-water sampling. The tap water faucet was externally cleaned with ethanol prior to sample collection, and the water was then left to flow freely for ∼30 s. Tap water samples for THM analysis were collected in falcon containers with oxidation preservative mixture, while the tap water samples used in the microbial analysis were collected in sterile polypropylene vials. The THM analysis was conducted according to the previously published methods by Charisiadis et al. (2015). All four THM species were measured in the collected tap water samples from the participating households: chloroform (TCM), bromodichloromethane (BDCM), dibromochloromethane (DBCM), and bromoform (TBM). The limits of detection (LOD) were 0.13 μg/L for TCM and DBCM, and 0.11 μg/L for BDCM and TBM.

Microbial analysis was conducted after the water was cultured onto selective media for the detection and enumeration of total coliforms, E. coli, Enterococcus spp., P. aeruginosa, and TVC at 22 and 37 °C. The methods used for the microbial analysis are presented the Supplementary Methods document.

All water samples were collected from the main faucet of the household used to satisfy their potable needs and it was directly connected to the municipality’s water supply.

Residual chlorine was measured with a portable photometer (MaxiDirect; Lovibond, Amesbury, England) in water directly collected from the tap using the DPD (N,N-diethyl-p-phenylenediamine) method.

Statistical analysis for the urban population study

Descriptive analysis of the questionnaire responses

Descriptive statistics (i.e., means and standard deviation (sd) for the continuous variables, and frequencies and percentage by category for the categorical variables) were calculated for the responses to the questionnaire. The descriptives were grouped by category of question (i.e., demographics and other background characteristics, drinking water habits and cleaning activities, lifestyle and behavioral indicators, healthcare services access, health status, life in the neighborhood) for the complete study population (n = 132).

Descriptive analysis of the water quality indicators

Descriptive statistics for the drinking-water THM and microbial counts were separately presented for the samples collected directly from the tap when no filter was attached (n = 119) from those 13 samples that were collected with a filter present. When THM values were below the LOD (n = 4 for TCM and n = 1 for TBM), they were imputed to LOD/2. The sum of all THM species (total THM) and the sum of the brominated species (BrTHM; i.e., BDCM, DBCM, and TBM) were calculated. For the statistical analysis of the microbial water quality we considered the presence or absence of colonies for E. coli and Enterococcus spp. instead of the absolute count number (EU Council, 1998). For the coliforms, P. aeruginosa, and TVC at 22 and 37 °C presence or absence were also considered. Specifically, for TVC at 22 and 37 °C a smaller number of samples was analyzed (n = 95) due to external contamination. For the THM concentrations the descriptives included: mean, sd, median, and percentiles, that is, 25th, 75th percentiles, and the range, that is, min and max, while for the microbial analysis the frequency of samples with at least one colony-forming unit (CFU) (and the percentage) was calculated. Water samples collected from households (n = 13) having permanently connected point of use filters in the main faucet were excluded from the main statistical analysis.

The results of the THM and microbial analyses were presented separately for the total THM, the E. coli and the Enterococci spp., as they are considered of higher priority compared to the single THM species and the other microbial indicators (e.g., coliforms).

Mapping of the water and the quality of life indicators

To evaluate how indicators pertaining to the water and quality of life are distributed in different quarters of Limassol, we mapped a selection of urban indicators by quarter. For the water quality indicators, the median levels of total THM, free chlorine, BrTHM per quarter were mapped, as well as, the percentage of samples with detectable levels of coliforms and TVC. From the urban questionnaire data, all indicators were mapped. However, for brevity, we present in the maps and discuss selected indicators per category for the following: (i) issues on access to health care services due to delays and financial constraints, (ii) life in the neighborhood, and (iii) two indicators from the category on access to green spaces to illustrate potential differences between quarters.

Exploratory environment-wide association study

We performed an exploratory environment-wide association study (EWAS) to agnostically synthesize knowledge and concurrently investigate possible associations of the measured water quality indicators and questionnaire-based socioeconomic, lifestyle, and behavioral factors. This exercise helped us demonstrate how an EWAS approach can be incorporated in analyses that are based upon the urban exposome framework. The outcomes used in the EWAS analysis were three self-reported health status outcomes: (i) general health status, (ii) diagnosed with any chronic disease, and (iii) diagnosed with any disease in the past year (“any disease the past year,” that is, asthma, diabetes, allergies, hypertension, cardiovascular or respiratory illnesses, depression, cancer, musculoskeletal problems) (Patel, Bhattacharya & Butte, 2010). The EWAS analysis included a correlation matrix between all variables, regression analysis, and a multi-omics-based approach to describe correlations between all the variables measured accounting for specific health outcomes.

The variables included in the EWAS approach were divided in the following “blocks/groups” to reflect different parameters of the human exposome, since the analysis was performed at the individual level:Block/group 1 (specific external domain): water THM levels, free chlorine

Block/group 2 (specific external and internal domain): drinking water habits (e.g., number of glasses of water consumed by source)

Block/group 3 (specific external and internal domain): household cleaning activities (e.g., dishwashing, mopping, bathroom cleaning)

Block/group 4 (general external domain): neighborhood quality of life (variables of “highest consensus”: heath care access, life in the neighborhood and green urban spaces)

Block/group 5 (internal domain): participant characteristics (e.g., age, sex, BMI)

Block/group 6 (internal domain): self-reported diseases in the past year (e.g., asthma, diabetes, allergies, hypertension, cardiovascular or respiratory illnesses, depression, cancer, or musculoskeletal problems).

In all categorical variables the “I don’t know/I don’t want to answer” responses were re-coded to “missing.” Then, scores were added per category (presented in Table S2) and used in statistical analysis when categorical variables could not be used (e.g., correlations).

In the preliminary correlation analysis, all variables (including the three outcomes) were used as continuous and the Spearman correlation coefficient was calculated without any transformations. The results were visualized with a correlation plot. Then, regression models were fitted. The variable for the general health status was used as continuous in linear regression whereas responses for self-reported chronic disease and any disease the past year were used as binary variables in logistic regression. The regression models were repeated after adjusting for age and sex. The continuous predictors were scaled and centered in all regression models. The p-values of all model parameters that were used for inference (i.e., excluding the intercept for the univariable models and excluding the intercept, age, and sex coefficients from the adjusted models) were summarized and corrected for Benjamini–Hoechberg false discovery rate (FDR). Only parameters with an FDR-corrected p-value < 0.10 separately applied to the univariate (n = 129 tests) and the adjusted models (n = 123 tests) were considered statistically significant.

In the last part of the EWAS analysis, we followed an approach used in multi-omics studies where the variables are grouped and partial least squared discriminatory analysis (block PLS-DA) is conducted to identify possible correlations between the variables of the different blocks accounting for an outcome (Rohart et al., 2017). In this analysis, the predictor variables were used as continuous and all outcomes were used as categorical. The correlations between the predictor variables in blocks were presented in circular plots (circo plots) where positive and negative Pearson pairwise connections are shown in the circle and lines indicate the levels of each variable within each outcome category.

All analyses (regressions and block PLS-DA) were performed for the three outcomes using all the blocks/groups as predictors except for the analysis for the outcome “any disease the past year.” This variable was created as the summary of the variables of block/group 6 (self-reported diseases the past year). Thus, in this analysis, the separate diseases of block/group 6 were not included due to their association with the outcome of “any disease the past year.”

All analyses were conducted in R 3.5.1 with RStudio 1.1.423 (RStudio Team, 2015; R Core Team, 2017). Input data, the output, scripts, and the questionnaires are available in the Supplementary Material. The packages used in the analysis are listed in Table S3.

Results

Perceptions study results

Approximately, 10 municipality technical officers were approached for interviews and to fill in the municipality questionnaire. In total, five interviews were conducted, and six questionnaires were administered. The importance of climate change and its effects was pointed out by all municipality officers during the face-to-face interviews and within the questionnaire responses. In rating the environmental concerns (one for very low and five for very high concern), water quality had the lowest score with increasing order of scoring for soil contamination < waste < general chemical exposures < noise < air pollution.

In total, 181 participants responded to the online questionnaire that was addressed to citizens and 134 (74%) filled in the complete questionnaire. A total of 91 of the respondents reported living in Limassol (35% males and 65% females) with a mean age of 35 years old (range:18–77 years old). The majority (81%) was born on Cyprus, and 13% were born in another EU country, the rest (6%) were born in a non-EU country. As expected from the distribution of the questionnaire among the staff of the university, most of the respondents (84%) were highly educated holding at least a Bachelor’s degree. Approximately half of the respondents were married (46%) with children (47%).

Residents of urban Limassol (n = 91) were mostly concerned about being severely exposed to air pollution and noise (Table 1). Water quality ranked low while air pollution and noise ranked high in the “severely exposed” category among all environmental exposures. Approximately 30% of the respondents reported that they were not exposed to water pollution or soil contamination, but at the same time an equally high proportion of respondents reported “don’t know,” suggesting inadequate knowledge about the drinking-water or soil quality in their city. With regards to water quality, 81% reported worrying about chemical exposures, and 41% reported they were exposed to chemicals daily.

Table 1 Perceptions about environmental exposures among the respondents from the city of Limassol (n = 91).

	Severely exposed	Somewhat exposed	Not exposed	Don’t know	
Noise (traffic, airplanes, factories, neighbours, animals, restaurants/bars/clubs)	38 (42%)	48 (53%)	5 (5%)	0 (0%)	
Air pollution (fine dust, grime, fume, ozone)	40 (44%)	40 (44%)	10 (11%)	1 (1%)	
Bad smells (industry, agriculture, waste)	15 (17%)	35 (38%)	39 (43%)	2 (2%)	
Water pollution (microbes/chemicals in drinking water)	8 (9%)	22 (24%)	31 (34%)	30 (33%)	
Soil contamination (eg., chemical waste dump)	5 (5%)	15 (17%)	32 (35%)	39 (43%)	

Among all urban respondents, only ∼30% reported the consumption of tap water, while 32% reported treating the water before consumption (filtering or cooking), and 39% mentioned not drinking the tap water at all. Also, the “highest in importance” concerns of the citizens about tap water quality were those associated with either chemicals (47%), or microbes (37%) and much less concern was expressed about the taste (10%).

Population study results

Background information and opinions

In total, 132 residents of the Limassol municipality answered the questionnaire and agreed to water collection from their households’ main tap. The distribution of the study participants by quarter and the population can be found in Table S1. The mean age was 45.6 years and the majority were females (62.1%). Most of the participants were born in Cyprus (n = 114 (86.4%)) living there for all their lives (Table 2; Table S4).

Table 2 Background characteristics of for the urban population study conducted in Limassol, Cyprus.

	Overall (n = 132)	
Age (mean (sd))	45.6 (13.2)	
Sex (%)	
 Females	82 (62.1)	
 Males	50 (37.9)	
 BMI (mean (sd))	26.25 (5.06)	
Marital status (%)	
 Cohabitating	8 (6.1)	
 Divorced	15 (11.4)	
 Married	82 (62.1)	
 Single	26 (19.7)	
 Widowed	1 (0.8)	
Education (%)	
 Primary school	2 (1.5)	
 Gymnasium	2 (1.5)	
 High School/Technical School	28 (21.4)	
 Non-university tertiary education	16 (12.2)	
 Post-secondary education	6 (4.6)	
 Bachelor’s degree (BSc/BA)	45 (34.4)	
 Master’s degree or doctorate (MSc/MA, PhD)	32 (24.5)	

Most of the study participants reported being in very good or good health condition (46% and 43%, respectively). However, 21% reported having a chronic disease and 57% reported at least one of the following health conditions during the past year: asthma, cardiovascular diseases, hypertension, diabetes, liver conditions, cancer, depression, or musculoskeletal problems (Table 3). In regard to access to health care, the main questions were about delays due to lack of transportation or long waiting lists (Table S5). Lack of transportation did not seem to be a major constraint to access health care centers among those that opted to answer; however, long waiting lists were reported by 11%. To the question about financial constraints for health care access, delays in dental care were most frequently mentioned (14%). Most participants (64%) reported living close to green spaces, but a total of 63% also reported that these green spaces were not well-maintained and there was a consensus for not using them. A summary of the responses about health care access, lifestyle, the quality of life in the neighborhood and other urban topics can be found in Table S5.

Table 3 Health status indicators assessed through the questionnaire among the 132 participants of the urban population study (Limassol, Cyprus).

	Overall (n = 132)	
General health assessment (%)	
 Very good	61 (46.2)	
 Good	57 (43.2)	
 So and so	12 (9.1)	
 Bad	1 (0.8)	
 Very bad	1 (0.8)	
Chronic disease (%)	
 Do not know	5 (3.8)	
 I don’t want to answer	3 (2.3)	
 No	97 (73.5)	
 Yes	27 (20.5)	
Any disease in the past year (%)	
 No	57 (43.2)	
 Yes	75 (56.8)	

Water quality indicators assessment of drinking water habits

The main chemical water quality indicators assessed were THM. Only 2% of the households’ tap water exceeded the THM parametric value (100 μg/L). Results conforming with the parametric values were also obtained for the microbial indicators monitored, that is, E.coli and Enterococci spp. All samples were within the parametric values (zero CFU per 100 mL), except for one household where Enterococci colonies were detected. Total coliforms were detected in 28 of the 132 households and P. aeruginosa counts were detected in five out of the 132 households (Tables 4 and 5).

Table 4 Chemical drinking water parameters analyzed in water samples collected in Limassol, Cyprus (2017) for n = 119 samples collected from faucets without point of use filter and for n = 13 samples collected from faucets with a point of use filter that could not be removed.

Samples collected from faucets without filter	
	n	Mean	Standard deviation	Median	25th percentile	75th percentile	Min	Max	
Regulated chemical parameters	
 Total THMs (μg/L)	119	30.5	32.22	15.2	6.99	50.83	3.17	210.5	
Non-regulated chemical parameters	
 Free chlorine (mg/L)	119	0.2	0.16	0.2	0.05	0.32	ND	0.7	
 TCM (μg/L)	119	2.1	2.24	1	0.47	3.16	0.07	13.3	
 BDCM (μg/L)	119	4.9	5.45	2	0.95	7.32	0.91	30.9	
 DBCM (μg/L)	119	9.1	11.18	2.4	1.18	15.52	1.07	66.2	
 TBM (μg/L)	119	14.4	14.63	7.7	3.94	22.6	0.06	100.3	
 BrTHMs (μg/L)	119	28.4	30.24	13.5	6.22	46.94	3.10	197.3	
Samples collected from faucets with filter	
	n	Mean	Standard deviation	Median	25th percentile	75th percentile	Min	Max	
Regulated chemical parameters	
 Total THMs (μg/L)	13	9.6	9.0	5.0	4.0	9.9	3.2	28.4	
Non-regulated chemical parameters	
 Free chlorine (mg/L)	13	0.1	0.1	0.1	0.0	0.1	ND	0.3	
 TCM (μg/L)	13	0.9	1.0	0.5	0.5	0.7	0.2	3.8	
 BDCM (μg/L)	13	1.7	1.7	1.0	0.9	1.1	0.9	5.8	
 DBCM (μg/L)	13	2.1	2.4	1.2	1.1	1.5	1.0	9.5	
 TBM (μg/L)	13	4.9	6.7	2.2	1.2	5.2	1.0	25.4	
 BrTHMs (μg/L)	13	8.7	8.5	4.7	3.5	9.4	2.9	28.0	

Table 5 Microbial drinking water parameters analyzed in water samples collected in Limassol, Cyprus (2017) for n = 119 samples collected from faucets without point of use filter and for n = 13 samples collected from faucets with a point of use filter that could not be removed.

Samples collected from faucets without filter	
	n	Samples with at least one CFU n (%)	
Regulated microbial parameters	
 E. coli (cfu per 100 mL)	119	0 (0)	
 Enterococcus spp. (cfu per 100 mL)	119	1 (0.8)	
Non-regulated microbial parameters	
 Coliforms (per 100 mL)	119	27 (22.7)	
 Pseudomonas aeruginosa (per 100 mL)	119	5 (4.2)	
 TVC at 22 °C (per one mL)	86	12 (14)	
 TVC at 37 °C (per one mL)	86	30 (34.9)	
Samples collected from faucets with filter	
	n	Samples with at least one CFU n (%)	
Regulated microbial parameters	
 E. coli (cfu per 100 mL)	13	0 (0)	
 Enterococcus spp. (cfu per 100 mL)	13	0 (0)	
Non-regulated microbial parameters	
 Coliforms (per 100 mL)	13	1 (7.7)	
 Pseudomonas aeruginosa (per 100 mL)	13	0 (0)	
 TVC at 22C (per one mL)	7	1 (14.3)	
 TVC at 37C (per one mL)	7	3 (42.9)	

The drinking-water consumption habits reported by the urban participants similarly reflected what was already observed in the perceptions study (presented earlier) where only 30% reported consuming tap water and most participants reported a preference for other sources (Table 6). The majority reported using tap water in general, however bottled water use was noted by most respondents. The more frequently reported single drinking water source was bottled water (30%) followed by tap water (22%). A comparable proportion of the participants reported the combined use of tap water and bottled water (Table 6). More than half of the study partipants reporting consuming less than one glass of water per day from the tap (median number of glasses consumed from tap was one glass/day) (Table 6).

Table 6 Self-reported choices of water sources by the study participants (n = 132) of the urban population study in Limassol, Cyprus.

Water sources	n (%)	
Bottled water	40 (30.5)	
Tap water	29 (22.1)	
Water from vending machines	4 (3.1)	
Spring water	3 (2.3)	
Tap water and bottled water	37 (28.2)	
Bottled water and spring water	3 (2.3)	
Bottled water and water from vending machines	2 (1.5)	
Tap water and other	2 (1.5)	
Tap water and water from vending machines	2 (1.5)	
Tap water and spring water	1 (0.8)	
Any three sources (tap water, bottled, water from vending machines, spring water or other)	5 (3.8)	
Number of glasses (about 250 mL) per day by source	n	Mean	SD	Median	
Tap water	130	2.52	3.6	0	
Bottled water	130	3.83	4.2	2.2	
Water from vending machines	130	0.54	2	0	
Spring water	130	0.33	1.3	0	
Other source	130	0.51	2.9	0	

Mapping of urban indicators of water and quality of life

The measured water quality indicators in each household were aggregated and mapped by quarter. For the chemical water parameters, the mapped median total THM and the brominated species (BrTHM) by quarter showed similar patterns (Fig. 2). As the BrTHM are a subset of the total THM, their median values by quarter were lower. The highest median THM values were observed in the quarters located between the beachfront and the northern quarter of Agia Fylaxi (near the center of the city). Mapping of free chlorine levels followed an opposite pattern to THM, that is, higher levels of free chlorine in the seaside quarters and slightly higher in the northern quarter (being closer to the main water treatment plant). The maximum median value of total THM was 63 μg/L and it was observed in the center of Limassol (quarter of Agios Georgios). In this quarter, the median free chlorine levels were below detection. The range of the median total THMs by quarter was 6–63 μg/L for the small quarters in the beachfront and behind the city port and the quarter of Agios Georgios. The variability of the median free chlorine levels was smaller, ranging from below detection to 0.4 mg/L in the quarters of Agia Zoni and Agia Trias (in the center and by the seafront, respectively). A map with the quarter names of municipality of Limassol is available in Supplementary Information (Fig. S1).

Figure 2 Maps of the median water total THM (A), BrTHM (B), and free chlorine (C) levels by quarter within the municipality of Limassol, Cyprus (2017).

With regards to the microbial parameters, as mentioned in the previous section, only one sample had detectable colonies of Enterococcus spp. while E. coli was not detected at all. Thus, we mapped only the percentage of samples found to be positive for coliforms or had detectable heterotrophic bacteria (TVC at 22 and 37 °C) (Fig. 3). In a one quarter, two of the three samples analyzed were positive for coliforms and, therefore, it had the highest percentage of samples with colonies. This quarter was geographically located in the zone with the highest median chlorine levels (0.3 mg/L) (Figs. 2 and 3).

Figure 3 Maps of the percentage of samples with detectable counts of the monitored microbial parameters, i.e. Coliforms (A), and total viable counts at 22 (B) and 37C (C), by quarter within the municipality of Limassol, Cyprus (2017).

Description and mapping of access to health care services, life in the neighborhood and use of green spaces

With regards to access to health care, most participants reported having issues with two major parameters, that is, long waiting lists and financial constraints to access dental care. From the respective maps (Fig. 4), participants living in the quarters of Katholiki, Agia Trias, Omonoia, and Agios Nektarios did not report any issues pertaining to access to health care. Whereas, other quarters such as Agios Ioannis/Arnaoutogeitonia, Agia Zoni, and Agios Nikolaos were more consistently in the “mid-range” with 20% participants reporting issues for both indicators. With regards to the question about having someone in the neighborhood to ask for help in emergencies, overall, only 25/132 participants opted for the answers “I don’t know” and “I disagree completely or probably” (Fig. 5). However, responses of strong agreement (“I completely agree” vs all the other options from “I probably agree” to “I completely disagree”) varied a lot across the quarters. For example, in Agios Spyridonas only 20% agreed that there is always someone to help them while in Agios Nikolaos 80% (Fig. 5). With regards to proximity and use of green space, most participants from all quarters reported that they agree living near to green space but they do not use it for activities (Fig. 6). This is evident in quarters by the seafront and the largest quarter of Agia Fylaxi/Panagia Evangelistria which is peripheral to the city the center and closer to less densely populated areas.

Figure 4 Percentage of study participants reporting constraints in access to health care, i.e. financial issue in access to dental care (A) and delays due to long waiting lists (B), by quarter of the Limassol municipality, Cyprus (2017).

Figure 5 Percentage of study participants agreeing with different statements about life in the neighborhood, i.e. on sharing the same values with the neighbors (A), and whether there is someone to help in the neighborhood (B), by quarter of the Limassol municipality, Cyprus (2017).

Figure 6 Maps by quarter of the percentage of study participants within the quarters of Limassol municipality, Cyprus (2017) agreeing that they live in close proximity to green space (A) and they do activities in the green space nearby (B).

Exploring environment-wide associations within the municipality of Limassol

A correlation plot between all variables used in the EWAS analysis (listed in Table S2) did not show any specific or unexpected patterns of correlation among the urban variables (Fig. 7). Notable correlations were observed among different variables of the same block/group. All THM compounds in drinking-water correlated well with each other and they were negatively correlated with free chlorine levels. Additionally, strong correlation was observed among the household cleaning variables (mopping, dishwashing and bathroom cleaning). The variables of the “cleaning” block/group (block/group 3) correlated also with certain health conditions, such as musculoskeletal problems (neck problems).

Figure 7 Correlation plot (Spearman correlation coefficient) for all the variables used in the environment-wide association exploratory analysis.

In the regression part of the EWAS analysis, a total of 129 predictors were summarized from the simple models and 123 predictors were used in the models adjusted for age and sex. These predictors include both parameters measured in water, that is, THM, and questionnaire responses (summarized in Table S2). In the models adjusted for age and sex, four parameters had an FDR-corrected p-value < 0.1, that is, financial issues in access to dental care, depression, hypertension, and asthma. Having encountered financial issues in access to dental care (n = 18, (14.1%) participants) and depression (n = 3, (2.8%) participants) were statistically significant negative predictors for better general health status, while higher odds of having a chronic disease were associated with hypertension and asthma (n = 19, (16.2%) and n = 10, (8.5%) participants, respectively). In the univariate regression, in addition to the parameters that were significant in the adjusted models, musculoskeletal problems (i.e., neck and back problem) were associated with higher odds of having a chronic disease but not with the outcome of general health status. The parameters ranked by the FDR adjusted p-value can be found in Table S6.

The results of the second EWAS analysis and the correlation between the variables used in the PLS-DA models were summarized in the circular plots (circo plots) of Fig. 8. Again, all three outcomes were used; however, the model for the chronic disease did not converge and thus the plot is not presented. For the other two outcomes the associations between the variables were not the same, as expected due to the difference in the outcomes between the two models. The THM variables correlated with each other and with the household cleaning activities (i.e., dishwashing, mopping, bathroom cleaning) as it was also shown in the simple correlation plots. The correlations were less in number when the outcome was “any disease the past year” compared to the correlations visualized in the circo plot of the “general health” as an outcome. Additionally, different levels of the predictor variables were noted depending on the outcome, as it can be seen by the lines on the outside perimeter of the circular plot. These lines, however, were not used for interpretation due to the study’s exploratory nature to avoid any misleading inferences.

Figure 8 Circular plots of the correlations between the variables used in the environmental-wide analysis by block/group of variables accounting for general health (A) and any disease the past year (B).

Discussion

In this proof of concept study of the urban exposome, we used an interdisciplinary approach to identify trends in perceptions about environmental exposures and how they correlated with the spatially-resolved drinking-water quality trends of chemical and microbial indicators for the municipality of Limassol during the hot season (summer). Quality of life indicators (e.g., access to green spaces, the life in the neighborhood and reasons for delay or financial constraint in access to health care) were mapped at the level of the quarter (smallest administrative unit). No clear disparities at the quarter level were observed for all neighborhood-based queries, but financial constraints, especially for dental care were noted. Additionally, we generated global linkages and correlations between the health status of urban participants and their environmental/lifestyle/behavioral exposures at the individual level. A total of 129 parameters from 36 variables either directly measured in water (water quality) or derived from the survey (quality of life in the city) were integrated using an exploratory, agnostic, exposome-based EWAS approach. The general health and chronic health status of the urban participants were significantly associated in regression analysis with different health outcomes (e.g., hypertension, asthma) and quality of life indicators (e.g., financial issues in access to dental care). Circular correlation plots were derived from 36 urban exposome variables which were divided in six groups and accounted for self-reported health indicators (general health and having any diseases the past year). This study is an application of the urban exposome framework and the first to generate a snapshot of the actual urban exposome of Limassol, Cyprus, with a focus on water and quality of life indicators.

Urbanization, migration and other drivers of societal changes in urban settings are shaping population health and quality of life in cities leading to interventions such as the large scale neighborhood renewal programs that aim to improve living conditions and quality of life in cities (Kramer et al., 2017). The impact of such neighborhood-based programs could be enhanced with the comprehensive assessment of environmental, economic, societal, and other health parameters in the city. Within the context of improving urban life, novel paradigms such as the exposome can be adapted to provide a more interdisciplinary approach in tackling urban problems. The urban exposome can be viewed simply as the totality of environmental exposures occurring in cities (Probst-Hensch, 2017; Robinson et al., 2018). However, this definition is rather centered on individuals and it does not include wider determinants of urban health. For example, a recent study described the human exposome in city-based cohorts and focused on specific exposure-effect associations in one or multiple cohorts being pooled together (Robinson et al., 2018). In this study, we propose a novel approach using the urban exposome as a study framework to describe systematically how broader urban characteristics can be evaluated with an interdisciplinary methodology. Thus, we have placed the city and its smaller areas as the measurable units in the center of this exposome approach. Moreover, we have used the perceptions and the urban population study as two sources of information that can be summarized to provide a snapshot of the urban exposome of Limassol focused on water and quality of life.

The case of water pollution and the provision of safe and clean water to urban dwellers presents one of the most challenging elements to be incorporated within the urban exposome framework. This might be due to the complex systems that drive water quality and water choices given also the aging water infrastructure in urban settings. These challenges are also evident in the literature and in studies of water quality indicators but their links with the exposome are scarce. To our knowledge, the literature in exposome approaches to evaluate water quality is limited and it has not focused on the urban environment or the general population. For example, using an environment-wide association study methodology, the pregnancy exposome study on the INMA-Sabadell Birth Cohort in Spain looked into modeled at the individual level water disinfection byproducts (THM), among a wider set of other exposures ranging from urinary markers of chemical exposures to air pollution and noise (Robinson et al., 2015). They showed that the three THM “classes” studied (total and brominated THM, and chloroform) were strongly correlated with each other, but not with the other environmental exposures (Robinson et al., 2015). Albouy-Llaty et al. (2016) explored the association between endocrine disruptors (atrazine metabolites and nitrate/atrazine mixture) in drinking water and preterm birth accounting for socioeconomic factors (deprivation index) in the Poitou-Charentes region of France (Albouy-Llaty et al., 2016). The exposure to atrazine and the nitrate/atrazine mixture at the individual level was inferred from routine community monitoring of water quality; preterm birth was found to be associated with the deprivation index at the level of the neighborhood but not with the exposure to atrazine metabolites. Exposure to atrazine (measured through the metabolite 2-hydroxyatrazine) was not found to be a significant risk factor for preterm birth when accounting for the socioeconomic status of the area (Albouy-Llaty et al., 2016).

This is the first application of the urban exposome framework that took place in a medium-sized European city, Limassol, Cyprus. We used an agnostic approach following the urban exposome framework and we described water quality indicators at the level of quarter (the smallest administrative area within a municipality), accounting for urban, general population characteristics and additionally including the opinions/perceptions of residents and municipality technical stakeholders. The analysis included a suite of water quality aspects (chemical and microbial) which belong to the internal domain of the urban exposome and the specific external domain of the human exposome, and stakeholders’ opinions about environmental issues (specific external urban exposome and general external human exposome). Moreover, quality of life was assessed through citizens’ answers about access to healthcare and green spaces, and it was included along with lifestyle/behavior, and demographics in the EWAS analysis. This analysis is part of developing the urban exposome profile of Limassol and provides a snapshot of the state of the city, which will be combined with analysis of routinely collected data.

As mentioned before, our study aimed to provide a methodological approach in a relatively small city. Therefore, the generated associations of different participant characteristics and quality of life indicators with general health or with having a chronic disease in the EWAS study should be interpreted with caution due to the lack of adequate statistical power. In our case, the EWAS analysis is limited by the data at hand but, it is more of exploratory nature, as it is a part of this proof of concept study. Previous EWAS-exposome studies have been broader, with larger sample sizes, and, thus, more power (Patel & Manrai, 2014).

One observation that stood out in the correlation analysis is the negative association of water THM with cleaning activities. Previous studies have shown positive correlations between cleaning activities and urinary THM levels, and not between water THM levels and cleaning activities (Charisiadis et al., 2014). However, our observation, taking into consideration previous work on the exposure assessment to disinfection byproducts (Gängler et al., 2018), indicates the complexity of exposure assessment using environmental measurements as proxies of exposure, especially in EWAS studies. The inclusion of the perceptions study in developing the urban exposome profile of Limassol, with the use of a qualitative and quantitative approach allowed important community concerns about their urban life to surface. Air pollution was ranked as the most significant concern among the study respondents. Besides the general interest of air pollution and its health effects, Cyprus experiences frequent dust storms (World Health Organization, 2018; Achilleos et al., 2014). These events have probably triggered a specific concern among the population making air pollution the most frequently reported environmental concern in our study. Overall, the results presented here, should be interpreted with caution because of limitations in the study design (i.e., the cross-sectional design) and the small sample size or possible sampling bias (especially in the perceptions study).

This proof-of-concept study aimed to showcase the utility of the urban exposome framework in a urban study setting extending the continuum of the human exposome concept, thus, providing the methodological background for future studies. We were able to demonstrate the use of different tools in an integrative and interdisciplinary approach to capture the baseline urban exposome of Limassol municipality in this case. The same approach can be generalized. For example, if applied to other urban dwellings with bigger sample, it will allow us to scale up the application of such integrative and multidisciplinary protocols and will allow for the wider transfer and generalizability of the results. Future studies should also incorporate a more comprehensive assessment of urban quality of life (Serag El Din et al., 2013). In this analysis, the indicators of quality of life were limited to life in the neighborhood or the use of green space and the questionnaires were based on indicators previously assessed in larger urban studies which had not included Cypriot cities (European Urban Health Indications System Part 2, 2012; Pope et al., 2016). Additional information about social life is available from this study population and it will be incorporated the urban exposome of Limassol. Next steps will incorporate the analysis of routinely collected data of the registries and human biomonitoring analyses in a more complete characterization of the urban exposome in Limassol city. Moreover, given the “stakeholders” assessment of environmental problems, we have moved on with air quality measurements throughout the municipality, which were conducted in the spring of 2018 while we explore how the routinely collected data from the one governmental station for air quality measurements can be also included. This study shows how the goal of developing the urban exposome framework can be achieved by using all the available information in a real-time assessment of urban health and provide a tool for decision-making to stakeholders.

Our analysis did not raise any specific concerns about the quality of tap drinking water at the urban quarter scale of Limassol city using both chemical and microbial indicators. It was shown however, that residents do not trust, in general, the tap water and opt for using bottled water or water from other sources such as the “vending machine” water (groundwater from mountainous wells) which is very common practice in Cyprus. The cost of tap water is lower compared to bottled or “vending machine” water, and thus not using tap water may pose additional burden to household budgets, as it has been shown in other studies (Massoud et al., 2013). Mapping the cost of water by quarter would likely be informative about the existence or not of differences within Limassol in the economic burden of water consumption. Urban green space was particularly noted for being close to participants’ households; however, limited use of such green spaces (e.g., parks) was reported. Both aspects of access to health care and the use of green spaces within Limassol, were studied for the first time. Our approach could form the basis for future targeted and more integrative urban studies on these topics.

In general, our first results on the application of the urban exposome are promising and in need for verification and expansion. First and foremost, given the different habits of the citizens, exposome studies need to be more inclusive in the assessment of different water sources. Besides the inclusion of standard water testing parameters, future studies should address participant perceptions which are linked to behaviors, and, thus, exposures. Developing the health profile of a city in urban exposome terms and integrating different approaches comes with several limitations. Spatio-temporal considerations should be accounted for in the dynamic nature of urban exposome profiling. Although enough to estimate the background levels of population exposures to chemicals, the relatively small sample size might have not allowed us to capture spatial differences of the indicators measured within the smaller city administrative units. Additionally, the lack of biomonitoring data on water-related exposures (i.e., to disinfection byproducts in urine) have hindered the full deployment of EWAS tool capabilities. However, the availability of urine biospecimen for this survey will allow us to use biomonitoring and untargeted metabolomics tools in a follow-up manuscript.

Conclusions

Developing sustainable city health profiles with the aid of the urban exposome framework is a novel approach, yet, far from being simple or reductionistic in approach. It demands a comprehensive characterization of relevant indicators ranging from drinking water quality to health perceptions and opinions collected from the general population and technical stakeholders, etc. The urban exposome framework and its application will pave the way for developing the next innovative solutions and public health interventions for the city. This proof of concept case study of the urban exposome in Limassol, Cyprus demonstrates the feasibility of using novel exposome approaches in studying the city and its smaller within-city areas (quarters) as the units of reference. Within this context, the absolute water quality indicators, city residents,’ and other stakeholders’ opinions need to be integrated and expanded along with exposomic profiles, such as metabolomics or other -omics platforms and human biomonitoring protocols.

Thus, we need to develop specific urban exposome studies where city-specific characteristics and within-city interactions and networks, can be used to redefine city health profiles. Evidence-based and city-specific studies will help authorities reach informed decisions about everyday life, about city infrastructure changes and their effects on urban health and personal exposures. This will help the interpretation of inter-city difference and will allow the timely evaluation of within-city challenges.

Supplemental Information

Supplemental Information 1 Map of the quarters of the Limassol municipality as they are used in the analysis.

Click here for additional data file.

Supplemental Information 2 Notes on the methodology used for the microbial analysis.

Click here for additional data file.

Supplemental Information 3 Population by quarter, sample size estimation and number of participants.

Click here for additional data file.

Supplemental Information 4 Variables from the urban population study questionnaire used in the environment-wide association study (EWAS) analysis. Part A of the table lists the outcomes, and Part B the predictors by block/group.

Click here for additional data file.

Supplemental Information 5 R packages used in the data analysis.

Click here for additional data file.

Supplemental Information 6 Additional characteristics of the study population based on the questionnaire responses of the urban population study.

Click here for additional data file.

Supplemental Information 7 Summary of the responses to different questions relating to health care access, lifestyle and quality of life in the neighborhood.

Click here for additional data file.

Supplemental Information 8 Parameters from the univariate models ranked by FDR adjusted p-value. In the categorical outcomes (i.e. chronic disease and any disease the past year, noted as “ChronicDisease” and “Disease12M” in the column “Outcome”) the estimate is the odds ratio.

Click here for additional data file.

Supplemental Information 9 The folder includes the questionnaires, datasets, output, and scripts used in the analysis.

Click here for additional data file.

Supplemental Information 10 STROBE Checklist for the urban population (cross-sectional) study.

Click here for additional data file.

We would like to thank the Municipality of Limassol and all the study participants. Special thanks to Ms. Andriana Till for her contribution to participant recruitment and to Dr. Stephanie Gaengler for the fruitful discussions during data analysis. We would also like to express our gratitude to Drs. Athos Agapiou and Diofantos Hadjimitsis for sharing the map templates for the quarters of Limassol.

Additional Information and Declarations

Competing Interests

Author Contributions

Human Ethics

Data Availability

The authors declare that they have no competing interests.

Xanthi D. Andrianou conceived and designed the experiments, performed the experiments, analyzed the data, contributed reagents/materials/analysis tools, prepared figures and/or tables, authored or reviewed drafts of the paper, approved the final draft.

Chava van der Lek performed the experiments, analyzed the data, prepared figures and/or tables, approved the final draft.

Pantelis Charisiadis performed the experiments, approved the final draft.

Solomon Ioannou performed the experiments, approved the final draft.

Kalliopi N. Fotopoulou performed the experiments, approved the final draft.

Zoe Papapanagiotou performed the experiments, approved the final draft.

George Botsaris performed the experiments, approved the final draft.

Carijn Beumer contributed reagents/materials/analysis tools, approved the final draft.

Konstantinos C. Makris conceived and designed the experiments, analyzed the data, contributed reagents/materials/analysis tools, authored or reviewed drafts of the paper, approved the final draft.

The following information was supplied relating to ethical approvals (i.e., approving body and any reference numbers):

The study was approved by the National Bioethics Committee of Cyprus (decision number: 2017/23).

The following information was supplied regarding data availability:

Raw data are available in the Supplemental Files.

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
