# Peer review of "Application of the urban exposome framework using drinking water and quality of life indicators: a proof-of-concept study in Limassol, Cyprus"

_PeerJ, doi:10.7717/peerj.6851_

## Round 0.1 · original submission · Major Revisions

The manuscript has been reviewed and requires substantial revisions. Please provide a point by point response that identifies each issue raised by each reviewer and how that issue was addressed in the revised manuscript.

Reviewer 1 ·

Basic reporting

The authors have presented a good background on this topic. The language use is professional, and the paper is structured conform journal guideline.

Experimental design

The manuscript is within the journal’s scope. It has well-defined research questions considering the current knowledge gap in the literature, followed by a rigorous investigation. Methods are well described.

Validity of the findings

Data and statistical analysis are robust and controlled.

Additional comments

Authors of this manuscript have presented a study concerning the application of the urban exposome framework using drinking water and quality of life indicators. The authors have found that urban drinking-water quality using conventional indicators of chemical (disinfection byproducts-trihalomethanes) and microbial (coliforms, E. coli, and Enterococci) quality did not raise
particular concerns. Also, The general health and chronic health status of the urban participants were not significantly (all >FDR corrected p-value of 0.1) associated with any urban life variables. The topic is interesting, an important issue and generally well written, well structured and contributes to the existing knowledge. However, there are still some occasional grammar errors through the manuscript especially the article ‘’the’’, ‘’a’’ and ‘’an’’ is missing in many places, please make a spellchecking.

The results and discussion section needs further improvement, compare your findings with the other author's conclusions.
• In general, the manuscript needs to revise again concerning technical and grammar errors.
• Please provide more deep discussion about your results, compare your findings with the other author findings.
• Please clearly state the novelty of this work.
• Please check the reference style, some of the references are not according to the journal style, especially the journals abbreviations.
• As said, it could be a failure to comprehend the main argument; and lightening the discussion would enhance comprehensibility and make the authors’ point clearer and stronger.
• Therefore, the reviewer recommends to further improve the manuscript before accepting it for publication.

Concluding Remarks
The work presented in this manuscript is an interesting topic, it needs some more efforts to improve it further. Reviewer recommend minor revision of this manuscript and publishing it only after specific improvement of the current version are made.

·

Basic reporting

This paper is generally well written and well-organized. As a ‘proof of concept’ paper, it is not meant to draw strong conclusions but rather to demonstrate the process the study went through as well as show ways that similar studies could be done in the future with more details and more specific goals and outcomes. In this sense, the paper does a good job of demonstrating that quality of life indicators can be paired with qualitative assessment/perceptions from study participants to map potential relationships between quality of life and both internal and external factors.

Despite an overall fit-for-purpose design, for a proof-of-concept study, there are some opportunities for improvement as follows (and discussed in more detail in the remaining sections)

1. In the qualitative perceptions study, it is often not clear from the paper what specifically was assessed and therefore what to think of the associations between that perception and other perceptions or qualitative measurements. Especially for relationships that are specifically discussed in the paper, it should be clear what was measured based upon information in the text and main tables. This is described in more detail in the line-by-line comments to the author.

2. There are several statements that are unclear, especially in the introduction and the materials/methods section. The paper is generally in clear, understandable English, with a handful of unclear statements. I have mentioned each of these statements in the line-by-line comments for the author.

3. As described in the paper, the “urban exposome can be defined as the totality of indicators (quantitative and qualitative) that shape the quality of life and health of urban populations (Andrianou & Makris, 2018)”. Reality dictates that only some subset of relevant items can be measured as indicators, and to be useful in this type of analysis they will need to have some known public health or quality-of-life link in the quantities seen. There’s no major issue with the methodology of measuring of THMs (minor comments are included later in this review), but I am struggling to understand the use of variation of low levels of THMs as an indicator for public health or quality of life. The paper makes a presumption that there is a link between say, the difference between 20 ug/l and 50 ug/l and some health outcome. WHO does have established recommended standards for each of the four major THMs individually and an equation as a suggestion for an additive measure for potential regulation without a single additive value (see https://www.who.int/water_sanitation_health/dwq/chemicals/THM200605.pdf). The EU also has a parametric TTHM value of 100 ug/l, and US EPA uses 80 ug/l with a locational running annual average. The TTHMs measured in this study are a single point in time, whereas the WHO and US EPA values are meant for chronic exposure. Within the WHO guideline, page 41 states “… neither clear evidence of a threshold nor a dose-response pattern of increasing risk with increasing concentration of THMs has been found (Reif et al, 2000).” There is considerable concern around THMs, including both regulatory and best practice literature on the reduction of THMs, but at the same time there is also a lot that is not known as well as some controversy around THMs. These unknowns and controversy are not mentioned. Therefore, I struggle to understand how the evaluation of differences in values below regulatory thresholds contributes to understanding of the exposome in instances where there is a lack of clear health-based values. Of 132 samples, all but 3 were below the EU value. I did not attempt to measure against the WHO values but suspect new or possibly none are above those guidelines. I recommend reading Cotruvo & Amato (2018) at https://doi.org/10/1002/awwa.1210 for a history of the use (and potential inappropriate causation) of THMs and certain health endpoints (although this article is not itself peer reviewed, it does link to many peer-reviewed, regulatory, and other authoritative matters on the subject that can be used for background). At a minimum, clearly stating why THMs were chosen considering not having a clear dose-response relationship is vital to demonstrating the value of this study and recognizing the controversy over the value of comparisons of THMs at below-threshold values is essential to the value of this work. One possible solution is to state that there is no intended health endpoint (especially at values less than the regulatory thresholds). Another is to code the TTHM results as above or below the regulatory limit and study the relationships with other factors as an ordinal variable (above/below) as opposed to an interval variable. Explanation as to the selection of THMs, acknowledgement of the unknowns and controversies will add more credibility to the paper. Variations in THMs does serve as a good proxy for water age in chlorinated systems, so to the extent that this is the intent it should be clearly stated. I recognize that being a proof-of -concept study this isn’t as important as it would be as if specific conclusions about the area were being derived, but regardless this should be addressed to avoid unnecessary controversy and focus the paper on the main findings.

Experimental design

As a proof of concept study, the selection of respondents and other controls are not as vital to the validity of the findings. The distribution of the perceptions survey was not very random (much of the activity was through a university, which lends itself to more highly-educated persons than average (and there may be other ways in which they aren’t necessarily representative of the population at large. However, since the paper’s conclusions are not contingent on a representative sample, this is not a fatal flaw (although mentioning this in the text would improve transparency). There are minor comments within the comments to authors.

Validity of the findings

This paper is a proof of concept study. It demonstrated the collection and correlation of data for the purposes of an “urban exposome framework using drinking water and quality of life indicators” as described in the title. With acknowledgement of the limitations of the data, the findings (which are essentially that the process is feasible) are appropriate.

The main non-process finding is that there’s a correlation between THMs and cleaning activities. It isn’t clear what cleaning activities specifically measures, and no plausible connection between the two is even mentioned. This finding requires more explanation.

Additional comments

- Overall, the data provided looks to be helpful and supports the paper.
- Overall, the paper appears well written and organized.
- Lines 50-51: It is not obvious why increasing city populations would make access to safe and affordable water more important than it would have been with a smaller population. It would seem that access to safe and affordable water was always (and will always) be extremely important.
- Line 65-68: The sentence starting with “to present how the urban….” Is unclear, especially the phrase “of the urban exposome framework that often relies upon a hypothesis-generating scope, instead”. Please clarify, and possibly split the sentence into two or more.
- Lines 72-73: The phrase “half of the urban population of Limassol resides within the municipality of Limassol” is unclear as it seems to refer to itself (half of area A lives in area A). Reword for clarity.
- Lines 119-123: This is the area that indicates that the perceptions survey is strongly weighted to the university, as it was “distributed via email among staff”. It is also unclear the method of reaching the public through “mailing lists”. Assuming these were email lists, generally such lists are for a specific interest or purpose. It would be helpful to know more about the distribution forum.
- Line 126: The methods section only states that participants “were recruited” but doesn’t clearly state now they were recruited.
- Lines 128-129: Since urine samples were not used in the study, why are they mentioned? If the reference is kept, please define “biomonitoring purposes”
- Lines 133-134: Recommend changing “small in area and population size quarters” with “Quarters that are small in area and population” for clarity.
- Line 137: Isn’t “street” and “household” redundant, in that if only one participant can be chosen in a street, doesn’t that automatically preclude two in a household? Also, what constitutes a “street”? What is a road crosses multiple quarters?
- Line 148: The beginning of the “water sampling and analysis” would be a good place to put in some discussion on how these indicators were chosen, what they mean, etc.
- Line 152: Was there a known minimum or maximum stagnation time? Since both THMs and chlorine residual are good proxies for water age, this value is potentially important. Was any information obtained about whether the fixture used was or was not used frequently in previous months? Was anything about building complexity (a, whether the sample comes from a single-family home, a small apartment building, a large apartment building, or something else)
- Lines 178-180: The statement here about samples with filters present here is confusing, as lines 165-166 state that those with filters were removed from the analysis.
- Line 198: “BrTHM” is used without introducing what it is and why it matters.
- Line 275: the statement that “most of the respondents (84%) were highly educated” provides evidence for my previous comment that the sampling methodology favored the university.
- Line 278-279: One cannot be “severely exposed to air quality”. I think this is supposed to be “severely exposed to poor air quality”.
- Line 303: Are the listed health conditions all of the ones that count as “chronic diseases”, is there a list elsewhere, or is it open-ended?
- Line 310 “consensus not using them” is unclear – is it meant to be a “consensus for not using them” or something else?
- Line 324: There are a handful of edits in the document that appear to be in track changes, such as the change on this line from “to” to “for”. Please remove all track change edits from the paper (accepting or rejecting as appropriate).
- Line 335: There’s a “Error1 Reference source not found” error on this page, likely for a broken reference link or maybe a link to a table/figure?
- Lines 379-381: The statement “besides for THM; for example, strong correlation was observed between household cleaning variables… and certain health conditions”. The sentence structure makes it sound like the correlation is between cleaning variables and health conditions. However, from elsewhere in the paper, the correlation is discussed as between THMs and other indicators (health, for example). Please clarify.
- The sentence “the correlation analysis revealed the association of water THM with cleaning activities”. Cleaning activities is one of the perception items that is unclear. The supplemental files include number of days dishwashing, amongst other things, which is not clear because dishes alone are not really a cleaning of the household.
- Line 512-513: the statement that no specific concerns about the drinking water quality were found is very helpful for clarity, but also leads to the question of why THMs?
- Figure 1: The relationship between human and urban is not clear in the figure. The columns do not align with each other. In the “application” column, it isn’t clear whether or not the “applications” are meant to align with the top line items.

---

## Round 0.2 · accepted · Accept

The Revised manuscript has been rereviewed and all issues raised by reviewers have been adequately addressed.

Reviewer 1 ·

Basic reporting

There is no any scientific or technical issue.

Experimental design

There is no any scientific or technical issue. The experimental design is correctly made.

Validity of the findings

There is no any scientific or technical issue. The data and findings are valid and meaningful.

Additional comments

Authors of this manuscript have adequately addressed all comments and suggestions made by reviewers; I would like to congratulate them for the effort. This version of the manuscript is significantly improved, therefore I recommend publishing the manuscript as it is.

·

Basic reporting

This paper remains as well written and well-organized. The authors have adequately addressed my previous concerns in this area.

Experimental design

My previous concerns around participant selection have been adequately addressed by the authors. The remainder of the experimental design is also well-constructed and reported.

Validity of the findings

The only concerns I previously had around the validity of the findings have been appropriately addressed by the authors.

Additional comments

Previously reported general and line-item comments for the authors have been addressed in this revision. I have no additional suggestions for revision, and I believe the paper is in good shape to move forward towards publication.